



# AtChem, an open source box-model for the Master Chemical Mechanism

Roberto Sommariva[1,5], Sam Cox[2], Chris Martin[3,4], Kasia Borońska[4], Jenny Young[3], Peter Jimack[4], Michael J. Pilling[3], Vasileios N. Matthaios[5], Mike J. Newland[6], Marios Panagi[1], William J. Bloss[5], Paul S. Monks[1], and Andrew R. Rickard[6,7]

[1]Department of Chemistry, University of Leicester, Leicester, UK
[2]Research Software Engineering Team, University of Leicester, Leicester, UK
[3]School of Chemistry, University of Leeds, Leeds, UK
[4]School of Computing, University of Leeds, Leeds, UK
[5]School of Geography, Earth and Environmental Sciences, University of Birmingham, Birmingham, UK
[6]Wolfson Atmospheric Chemistry Laboratories, Department of Chemistry, University of York, York, UK
[7]National Centre for Atmospheric Science, University of York, York, UK

**Correspondence:** R. Sommariva (rob.sommariva@gmail.com)

**Abstract.** AtChem is an open source zero-dimensional box-model for atmospheric chemistry. Any general set of chemical reactions can be used with AtChem, but the model was designed specifically for use with the Master Chemical Mechanism (MCM, http://mcm.york.ac.uk/). AtChem was initially developed within the EUROCHAMP project as a web application (AtChem-online, https://atchem.leeds.ac.uk/webapp/) for modelling environmental chamber experiments; it was recently upgraded and

further developed into a standalone offline version (AtChem2) which allows the user to run complex and long simulations, such as those needed for modelling of intensive field campaigns, as well as to perform batch model runs for sensitivity studies. AtChem is installed, set up and configured using semi-automated scripts and simple text configuration files, making it easy to use even for non-experienced users. A key feature of AtChem is that it can easily be constrained to observational data which may have different timescales, thus retaining all the information contained in the observations. Implementation of a continuous

integration workflow, coupled with a comprehensive suite of tests and version control software, makes the AtChem codebase robust, reliable and traceable. The AtChem2 code and documentation are available at https://github.com/AtChem/, under the open source MIT license.

## 1   Introduction

Computational models play an integral role in the study of atmospheric chemistry, air quality and climate. The interpretation of

ambient measurements and of laboratory/environmental chamber experiments relies on chemical models, which, in turn, inform and direct the focus of field studies and of the experimental investigations of fundamental chemical and physical processes (Abbatt et al., 2014; Burkholder et al., 2017). Of particular importance to atmospheric chemistry are zero dimensional box-models: this type of model considers the chemical species within an air parcel to be uniformly distributed, so that all points within the box are equivalent, effectively reducing the model to a single, zero dimensional, point. This modelling approach is





useful because it allows the user to focus on the fast radical chemistry and to neglect, to a first approximation, the effects of physical and meteorological parameters.

Zero dimensional box-models have long been used to analyze ambient measurements and environmental chamber experiments. There is a natural mapping between a zero dimensional box-model and the static nature of a measurement site (Eisele et al., 1994; Carslaw et al., 1999; Emmerson et al., 2007; Elshorbany et al., 2009; Lu et al., 2012; Edwards et al., 2014; Brune

et al., 2016; Whalley et al., 2016) and of an environmental chamber (Carter, 1995; Bloss et al., 2005a; Metzger et al., 2008; Chen et al., 2015; Novelli et al., 2018). With some modifications, the same modelling approach can also be used to analyze ship-based (Brauers et al., 2001; Sommariva et al., 2009, 2011a) and aircraft-based (Chen et al., 2005; Ren et al., 2008; Sommariva et al., 2008, 2011b) observations, and to simulate the chemical evolution and photochemical processing of air masses (Derwent et al., 2003; Madronich, 2006; Roberts et al., 2007)).

The core of a zero dimensional box-model is the chemical mechanism, which describes the chemical system that is being modelled. At a mathematical level, the chemical mechanism is a system of coupled ordinary differential equations (ODE) in the form:

$$\frac{d\boldsymbol{y}}{dt} = f(t, \boldsymbol{y}), \qquad \boldsymbol{y}(t_0) = \boldsymbol{y_0} \tag{1}$$

where $\boldsymbol{y}$ is the vector of the concentrations of the chemical species in the mechanism and $t$ is time. The system of ODEs is

then solved versus time from the vector of the initial concentrations of each species ($\boldsymbol{y_0}$) using a numerical integrator. Atmospheric chemical mechanisms can be very large, requiring an efficient mathematical solver capable of dealing with hundreds or thousands of ordinary differential equations (i.e., chemical reactions).

One of the most widely used chemical mechanisms for atmospheric chemistry is the Master Chemical Mechanism (MCM, http://mcm.york.ac.uk/, previously at http://mcm.leeds.ac.uk/). The MCM is a near-explicit chemical mechanism which de-

scribes the gas-phase oxidation of 143 (in version 3.3.1) primary emitted Volatile Organic Compounds (VOC) to carbon dioxide ($CO_2$) and water ($H_2O$). The MCM was originally assembled to model ozone formation (Derwent et al., 1998, 2003) and has since been adopted by the atmospheric chemistry community for a wide variety of research applications, as well as for policy and education activities. The protocol used to assemble the MCM was described in Jenkin et al. (1997), and subsequently updated in Saunders et al. (2003); Jenkin et al. (2003); Bloss et al. (2005b); Jenkin et al. (2015). The MCM protocol is designed

to strike a balance between the need to preserve the complexity of the chemical system and the necessity to contain its size, in order to make it computationally efficient. For this reason, the MCM has often been used as a benchmark to evaluate and optimize more complicated or more simple chemical mechanisms (Emmerson and Evans, 2009; Chen et al., 2010; Jenkin et al., 2002, 2019) and to generate reduced chemical mechanisms for use in three dimensional chemical transport models, which need to be orders of magnitude smaller than the MCM, owing to the limitations of computational power (Jenkin et al., 2002, 2019).

This paper presents the AtChem box-model, developed with four main objectives as part of the EUROCHAMP project (https://www.eurochamp.org/), which coordinates the activities of environmental and atmospheric simulation chambers in Europe. The first objective was to create a free and user friendly model to facilitate the use of the Master Chemical Mechanism.





Although access to the MCM database is fairly simple – via the tools available on the MCM website – the chemical mechanism alone cannot be used directly and, therefore, the setup and configuration of a complete box-model may be difficult for a
non-experienced user. AtChem incorporates the chemical mechanism into a program that manages the initial conditions and the various inputs required, so that the ODE system can be integrated by a numerical solver, with the outputs made available to the user in a suitable format. Second, there is a need to keep the MCM updated to the latest developments and experimental studies. To this end, an easy to use model that allows the atmospheric chemistry community to quickly run simulations of their experiments and provide feedback to the MCM maintainers/developers is highly desirable. Third, box-models are very useful
tools for teaching and outreach. AtChem was initially developed as a web application, which is simple to use in a classroom (at university level) and can even be used to communicate with the general public, as well as for citizen science initiatives. Finally, there are increasing concerns in the scientific community about the sustainability, traceability and reproducibility of computational models (Ince et al., 2012; Shamir et al., 2013; Bonet et al., 2014). Scientific code is often developed by programmers who don't have a software engineering background and therefore it may lack strict adherence to language standards, use of
modern programming techniques, and sometimes even proper documentation, which may make it difficult to reproduce published model studies and results, a key aspect of the scientific process. Addressing all these issues requires well documented, open source code, rigorously tested, and consistent tracking and documentation of all changes.

AtChem was conceived with the above principles and objectives in mind: the code is free, open source and publicly available. Although AtChem was designed mainly as a tool to encourage the use of the MCM in atmospheric chemistry studies, it can be
easily adapted to model other chemical systems and to use different chemical mechanisms, as long as they are provided in the correct format. This paper is divided into two parts: Section 2 describes the AtChem model architecture, setup and configuration, while Section 3 demonstrates its use for modelling environmental chamber experiments and ambient measurements.

## 2 Description of the AtChem model

### 2.1 Model architecture

AtChem was initially developed as a web application to provide a modelling tool for laboratory and environmental chamber studies that could be used by both experienced and novice users, particularly within the EUROCHAMP community. The original version, which will be referred to as AtChem-online in this paper, is compiled and run on a dedicated web server and can be used with just a text editor, file compression software, a web browser and an internet connection. AtChem-online is accessible at https://atchem.leeds.ac.uk/webapp/, with a simple tutorial available at http://mcm.york.ac.uk/atchem/tutorial_
intro.htt: the user simply needs to provide the chemical mechanism, the configuration files and the model parameters via a web form. The model results are stored on the web server and can be downloaded as compressed zip files for further processing and analysis.

While relatively simple and easy to use, AtChem-online has a number of limitations, mostly related to its nature as a web application. It cannot be customised by a user beyond what the Web Interface allows and, more importantly, it cannot be used
for batch model runs – i.e., multiple runs of the same model with minor and/or incremental modifications, a modelling approach





which is very useful for sensitivity studies. Moreover, the models required for ambient measurements and field campaigns are often more complex than those required for environmental chambers and laboratory experiments and need to be run for longer periods of time (several hours or days). Such models can be computationally very expensive and are therefore difficult to run from a web server with limited resources.

AtChem2 was developed from AtChem-online to overcome these limitations. The aim of AtChem2 was to create an offline version of AtChem capable of running long simulations of computationally intensive models and to make it possible to run batch simulations. Although the codebase has been extensively reworked, the basic architectures of AtChem-online and AtChem2 are very similar (Fig. 1). The structure and functions of AtChem are organized in five independent components, plus the chemical mechanism which is provided externally (Sect. 2.2):

– Web Interface: graphical user interface of AtChem-online, accessible via a web browser. In AtChem2, which does not run as a web application, this component has been removed.

– Configuration Layer: initial conditions, model constraints, input and output variables, model and solver parameters.

– Processing Layer: conversion of the chemical mechanism into Fortran format, sum of organic peroxy radicals, parametrization of photolysis rates.

– Logic Layer: conversion of the chemical mechanism and model configuration into a system of coupled ODEs, boundary conditions of the ODE system.

– Mathematical Layer: interpolation of constrained variables, integration of the ODE system.

Most of the AtChem codebase is written in Fortran 90/95; Python and shell scripts are used in the Web Interface, the Processing Layer and the Configuration Layer. The source code of AtChem-online is available at https://atchem.leeds.ac.uk/
sources/, while the source code and the documentation of AtChem2 are available at https://github.com/AtChem/, under MIT license. AtChem2 can be installed on a Unix/Linux or macOS machine and requires the user to have an elementary knowledge of the Unix shell. Installation of AtChem2 (and of its dependencies) is semi-automated via a number of well documented scripts that require minimal input from the user. The compilation of AtChem2, which is also done via a script, creates an executable file which reads the configuration of the model at runtime from a directory chosen by the user. For both versions of AtChem, the
model configuration – including inputs, outputs and constrained variables – is set via simple text files, which can be modified with a normal text editor. In AtChem2 the configuration files are stored in a dedicated directory, while in AtChem-online they need to be uploaded (together with the chemical mechanism) to the web server.

The AtChem-online codebase (rev. 146) was the starting point for the development of AtChem2. Several parts of the code were modified: the web tools were removed, the code was reorganized in Fortran modules, thoroughly commented and par-
tially rewritten to fully conform to the Fortran 90/95 standard. An important addition to AtChem2 is the implementation of a continuous integration workflow for the development of the model coupled with an extensive suite of tests, which means that every change to the source code is automatically checked against previous model results before being accepted into the





codebase. In recent years, continuous integration and testing have become standard practice in the software industry, allowing programmers to quickly detect bugs and errors, to ensure that modifications to the code do not result in unintended behaviour, and to improve the overall quality of the code. The suite of tests in AtChem2 includes unit tests of individual model functions and complete model runs: it is designed to cover a significant percentage of the codebase ($\sim$90%) and a wide range of common model configurations. Together with the use of the open source version control software git (https://git-scm.com/), these modern software development practices make the AtChem2 model easy to maintain, robust and reliable, as well as fully traceable and reproducible.

## 2.2 Chemical mechanism

AtChem is designed to use the Master Chemical Mechanism (MCM) as its chemical mechanism. The entire MCM, or a subset of it, can be downloaded from the MCM website in a variety of formats using the online extraction tool (http://mcm.york.ac.uk/extract.htt). The current version of AtChem requires the chemical mechanism to be provided in a format compatible with the one used by FACSIMILE (Curtis and Sweetenham, 1987), a common commercial software for modelling the kinetics of chemical and physical systems (MCPA Software Ltd., UK). The advantage of this format to describe a chemical mechanism is that it is simple, and easy to read and modify. A chemical reaction is described with the following notation:

`% k : R1 + R2 = P1 + P2 ;`

where `k` is the rate coefficient, `R1` and `R2` are the reactants, `P1` and `P2` are the products.

The chemical mechanism file extracted from the MCM website does not need to be modified in order to be used in AtChem. A chemical mechanism different from the MCM can be used, provided that it is in the correct format and it follows the requirements of the MCM. In particular, the calculation of photolysis rates and the sum of organic peroxy radicals ($RO_2$) must be treated as described in the MCM protocol papers (Jenkin et al., 1997; Saunders et al., 2003). These aspects of the AtChem model are further discussed in Sect. 2.3 and Sect. 2.4.

In order to create the executable file, the chemical mechanism needs to be converted into a format readable by the Fortran compiler, a task performed by a series of Python and shell scripts during the build process (Sect. 2.5). In AtChem-online the conversion is done once the user has uploaded the chemical mechanism file (with the configuration files) to the web server via the Web Interface, while in AtChem2 the user simply needs to execute a shell script and give the name and path of the chemical mechanism file (Fig. 1). The chemical mechanism is the only part of the model that needs to be compiled with the Fortran source code: all the configuration files – inputs, outputs, constraints, model and solver parameters – are read into the model at runtime, meaning that changes in the model configuration do not require the model to be recompiled (Sect. 2.5).

## 2.3 Variables and constraints

AtChem, and the MCM, have three types of variables:

- Chemical Species: atoms and molecules in the chemical mechanism. The exceptions are $CO_2$ which, as an end product of VOC oxidation, is not considered by the MCM, and $H_2O$ which is an environment variable (see below); molecular





oxygen and nitrogen ($O_2$ and $N_2$) are treated as model parameters and their concentrations are calculated from temperature and pressure. A special chemical variable is RO2, the sum of all the organic peroxy radicals, which is calculated at runtime by the model using the complete list of organic peroxy radicals in the MCM. The sum of $RO_2$ is a key element of the MCM protocol – an approximation designed to reduce the number of peroxy radical self and cross reactions (Jenkin et al., 1997). The list of organic peroxy radicals can be empty if a mechanism other than the MCM is used, in which case

RO2 has a value of zero.

– Environment Variables: physical characteristics of the model, such as temperature, pressure and solar angles (sun declination, solar zenith angle). Water ($H_2O$), which can be calculated from relative humidity, is considered an environment variable, not a chemical species. Additional environment variables allow the user to apply a scaling factor to the photolysis rates (JFAC, Sect. 2.4) and to use specific parameters for ambient studies (e.g., boundary layer height) or for

environmental chamber experiments (e.g., chamber dilution, roof open/closed).

– Photolysis Rates: reaction rates of the photolysis reactions in the chemical mechanism. The treatment of photolysis rates in the model is described in detail in Sect. 2.4.

All chemical species, most environment variables and all the photolysis rates can be constrained to prescribed values, such as ambient or chamber measurements. When a variable is constrained, the solver is forced to use its value at each time step to

calculate the values of the other variables. The constrained data are stored as simple text files in the corresponding directories.

Constrained box-models are often used to study the chemical processes in a given location (e.g., where a field campaign has taken place) or in a chamber experiment. The rationale behind this modelling approach is that short lived reactive species are not significantly affected by atmospheric transport or other physical processes. Radical species – such as OH, $HO_2$, $RO_2$ and, under certain conditions, $NO_3$ (Brown et al., 2003; Sommariva et al., 2009) – have lifetimes between a few seconds to a few

minutes. Therefore, the *in-situ* concentrations of radicals can be calculated from the measured concentrations of longer-lived species and from the measurements of other parameters (photolysis rates, temperature, pressure, etc...). Hence, the ability of the model to reproduce the observations of radical species is an effective test of the description of atmospheric chemical processes in the model (Eisele et al., 1994; Carslaw et al., 1999).

The main problem of this modelling technique is that the datasets of constrained variables are often provided with different

time frequencies, depending on the instrument or analytical technique used for the measurement. Some species (e.g., $O_3$, NO, $NO_2$) are usually measured once every minute, while others (e.g., most VOC measured by gas chromatography) are typically measured once every 30-60 minutes. Additionally, data from some instruments may be missing for short periods of time, due to operational limitations, calibrations or instrument downtime. A common method to address this issue is to average the constraints to the lowest time frequency available (e.g., 30 minutes). However, this introduces significant uncertainties in the

model results and does not allow investigations of the short scale changes in atmospheric composition (Sonderfeld et al., 2016).

An alternative approach is to interpolate the model constraints to fill the gaps and compensate for the different timescales. In AtChem, each constraint is separately interpolated at runtime, using piecewise linear interpolation (piecewise constant interpolation is also available). The advantage of using an interpolation method is that setting up the model is easier and faster,





as there is no need to average the constrained data onto a single time base beforehand. More importantly, the constrained data
can be used with the original time frequency, thus retaining the important kinetic and mechanistic information that is lost by
averaging to the lowest time frequency (Sonderfeld et al., 2016). The disadvantage is that some assumptions are made about
the time evolution of the low frequency constraints, which may lead to serious errors if, for example, the gaps in the data are
large or the short term variability is high.

The impact of the frequency of the constrained data on the model results was investigated using an AtChem2 model con-
strained to the measurements of 32 chemical species, 18 photolysis rates and 4 environment variables. The frequencies of the
measurements are shown in Table 1. Three model scenarios were used: in all scenarios, methane and C2-C7 hydrocarbons
were averaged to 60 minutes, while C1-C4 oxygenated hydrocarbons, CO and $H_2$ were averaged to 15 minutes. In scenario
A, the photolysis rates, the environment variables, and the chemical species $O_3$, NO, $NO_2$, $SO_2$ were averaged to 15 minutes.
Scenario B was identical to scenario A, except the photolysis rates were not averaged but used with the original measurement
frequency (1 minute). Scenario C was identical to scenario B, except the environment variables and the chemical species $O_3$,
NO, $NO_2$, $SO_2$ were not averaged but used with the original measurement frequency (1 minute).

The model was run for 9 days, with a 12 hour spin-up period in order to get short lived intermediates into steady-state: as
explained above, AtChem interpolated the constrained data at runtime where necessary. The relative differences between the
modelled concentrations of a target species (e.g., OH or $HO_2$) in each scenario were calculated with Eq. 2:

$$\Delta X_i = \frac{X_i - X_A}{X_A} \tag{2}$$

where $X_i$ is the concentration of the target species in scenario $i$ and $X_A$ is the concentration of the target species in the
reference scenario (A). Scenario A was used as reference because averaging all measured data to 15 minutes is common
practice for constrained models; OH and $HO_2$ were chosen as target species because of their central role in this type of
modelling studies, as explained above. Figure 2 shows the diurnal distributions of the median relative differences, binned by
hour of the day, for the 9 days model run.

The model constrained to 1 minute photolysis rates (scenario B) calculated higher concentrations of OH and $HO_2$ (10-15%
in the morning and ∼5% in the afternoon) compared to the model constrained to 15 minute photolysis rates (scenario A).
Increasing the frequency of the chemical species $O_3$, NO, $NO_2$, $SO_2$ and of the environment variables (scenario C) resulted in
even larger changes in the calculated concentrations of OH and $HO_2$ at all times of the day, with variations of up to 20% for
OH and up to 15% for $HO_2$. In both scenarios B and C, the differences in the calculated radical concentrations were higher –
up to 40% relative to scenario A – during sunrise and sunset than during the rest of the day (Fig. 2). These periods are critical
for a model from a chemical and mathematical point of view, because they correspond to the sharp changes in the atmospheric
chemical processes caused by the photochemical reactions starting and stopping (respectively). These discontinuities typically
results in increased stiffnes of the ODE system (Sect. 2.6) leading to larger uncertainties in the calculations.

Figure 2 shows that the frequency of the constrained variables has a significant effect on the model results, especially during
sunrise and sunset. The interpolation of constraints allows the model to use as many high frequency data as are available,





resulting in more precise, if not more accurate, model results. It must be noted that the use of high frequency data as model constraints has the downside of slowing down the integration of the model. For example, the model runtime for scenario C is approximately 20-30% longer than for scenario A. It is up to the user to decide on the balance between model precision and
model runtime, depending on the objectives of the modelling work and on the available computing resources.

## 2.4  Photolysis rates

AtChem implements the parametrization used by the MCM to calculate the photolysis rates of the appropriate chemical species under clear sky conditions (Jenkin et al., 1997; Saunders et al., 2003). Each photolysis rate ($j$) is calculated with Eq. 3:

$$j = l * (cos(\text{SZA}))^m \times e^{(-n \times sec(\text{SZA}))} * \tau \tag{3}$$

where $l$, $m$ and $n$ are empirical parameters, SZA is the solar zenith angle and $\tau$ is a transmission factor. The empirical parameters are obtained, for each version of the MCM, by fitting Eq. 3 to the output of a two stream isotropic scattering model, which incorporates the appropriate photolysis cross-sections and quantum yields (Jenkin et al., 1997; Saunders et al., 2003). The transmission factor $\tau$ can be used to account for the loss of natural or artificial light in some environmental chambers caused, for example, by the transmittance of the chamber walls (by default, $\tau = 1$). The user can customise the photolysis rates
parametrization by providing an alternative file to replace the default values of $l$, $m$, $n$ and $\tau$. The solar zenith angle (SZA) is calculated by AtChem from latitude, longitude, day of the year, time of the day and sun declination according to Madronich (1993). The photolysis rates can also be set to a constant value or constrained to measured data: the flowchart in Fig. 3 shows how AtChem combines constant, calculated and constrained photolysis rates, depending on the model configuration.
A correction factor (`JFAC`) can be used to account for the difference between the photolysis rates, which are calculated by
the model under clear sky conditions, and the measured photolysis rates, which are affected by other environmental factors (e.g., clouds and aerosol). A measured photolysis rate is used as a reference to calculate `JFAC` using Eq. 4:

$$\text{JFAC} = \frac{j_{meas}}{j_{calc}} \tag{4}$$

where $j_{meas}$ and $j_{calc}$ are the measured and calculated (with the MCM parametrization) photolysis rates for the reference species, usually $NO_2$. `JFAC`, which can also be provided by the user and constrained as an environment variable, is then
applied to the other calculated photolysis rates, as shown in Fig. 3.
Figure 4 shows a comparison between the photolysis rates calculated with the MCM parametrization and measurements of $j(NO_2)$ and $j(O^1D)$ made in different seasons in Boulder, CO (USA). The model correctly calculates the solar angles (sun declination, solar zenith angle, local hour angle and equation of time) and the appropriate diurnal profiles defined by the photolysis cross-section wavelength thresholds, as demonstrated by the correct timing of sunrise, midday and sunset (Figure 4).
The calculated values of sun declination and solar zenith angle for the 5 years period 2004-2009 were also double-checked with the online solar calculator of the National Oceanic and Atmospheric Administration (NOAA, https://www.esrl.noaa.gov/gmd/grad/solcalc/) and the agreement between AtChem and the NOAA tool was within 1%.





On average, the model underestimated the measurements by 25-30% in all seasons, with slightly better agreement (within

20%) in autumn. The discrepancies between the modelled and the measured photolysis rates may be due to several factors: in

particular, the two stream isotropic scattering model used to derive the empirical parameters in Eq. 3 is run for an altitude of

500 m and a latitude of 45° N on July 1 (as described in Jenkin et al. (1997)), while the measurements shown in Fig. 4 were

taken at an altitude of ∼1700 m and a latitude of 40° N in different seasons and years (between 2004 and 2009). Additionally,

the model assumes clear sky and ideal environmental conditions, which is often not the case during ambient measurements.

The discrepancies between the model and the measurements thus highlight the importance of using measured photolysis rates

(if available) and of using JFAC to correct the calculated photolysis rates, as explained above.

## 2.5  Model configuration

The typical workflow for AtChem2 is shown in Fig. 5: the user downloads the chemical mechanism from the MCM website,

prepares the configuration files and chooses the model parameters. For AtChem-online a few extra steps are required, as the user

has to upload the model configuration and data to the web server via the Web Interface (Fig. 1). The configuration of AtChem

sets the initial conditions, the list of constrained variables, the model start/stop date and time, the latitude and longitude, and

the required model outputs. All the model configuration information and data are provided to AtChem in the form of simple

text files, which can be prepared and edited with a normal text editor, thus simplifying the setup of the model and eliminating

the need to modify the Fortran source code.

Compilation of the AtChem model is done via a series of Python and shell scripts which link together the Fortran source

code and the chemical mechanism – after conversion to a Fortran compatible format, as explained in Sect. 2.2 – to create an

executable file, called atchem (for AtChem-online) or atchem2 (for AtChem2). The compilation process is performed with

a build script, which requires only a basic knowledge of the Unix command-line: the user has to pass to the build script the paths

to the chemical mechanism file, to the configuration directory and to the model constraints. Since the model configuration and

constraints are read by the executable at runtime, there is no need to compile the model more than once, unless the chemical

mechanism or parts of the source code are modified by the user (Fig. 5), which makes it quick and easy to set up batch model

runs. In the case of AtChem-online, compilation is automatically done on the web server when the model run is started: batch

model runs are not possible, because the chemical mechanism, the configuration files and the model constraints have to be

uploaded via the Web Interface and the model has to be recompiled every time it is executed, regardless of the changes that the

user has made.

## 275  2.6  Integration and output

An atmospheric chemistry model is essentially a system of coupled ODEs that needs to be solved versus time for a given set

of boundary conditions (Sect. 1). AtChem interpolates between the data points of the constrained variables, as explained in

Sect. 2.3: the chemical species, the photolysis rates and the environment variables are evaluated by the solver when required

and each is interpolated individually during the integration of the ODE system.





AtChem uses the CVODE library, which is part of the SUite of Nonlinear and DIfferential/ALgebraic Equation Solvers (SUNDIALS, Hindmarsh et al. (2005)) to integrate the system of differential equations; SUNDIALS is open source, under BSD 3-Clause license, and is available at https://computation.llnl.gov/projects/sundials/. Atmospheric chemical models are usually very stiff: this means they have at least one rapidly-damped mode, corresponding to the short atmospheric lifetimes of some chemical species (of the order of seconds to minutes for the OH, $HO_2$ and $RO_2$ radicals) relative to the timescales of the

full system (of the order of hours to months). The disparity in timescales results in the stiffness of the underlying ODE system. CVODE uses a multi-step method with variable step-size and variable order to solve this type of stiff system. The solver type, preconditioner, and other solver settings can be tuned by the user, although the default settings should be good enough for most atmospheric chemistry box-models.

    AtChem outputs the concentrations of the chemical species, the values of the environment variables, the reaction rates, the

photolysis rates, and the model diagnostic variables. The Jacobian matrix can also be output, if required. Because of the large number of chemical species in the MCM (over 7000), the output of the calculated concentrations is currently limited to 100 chemical species selected by the user in the model configuration (Sect. 2.5), although this number can be changed by modifying the Fortran source code.

    Reaction rates ($\mathtt{k} \times [\mathtt{R1}] \times [\mathtt{R2}]$, for the generic reaction $\mathtt{R1 + R2 \rightarrow P1 + P2}$) are output for all reactions in the chemical

mechanism at a frequency chosen by the user in the model configuration. In addition, the model can calculate and output the rate of production and destruction for a selected number of species of particular interest. Rate of production/destruction analyses (ROPA/RODA) of short lived reactive species are very useful to investigate the chemical budgets and fluxes of species of particular interest, such as the OH, $HO_2$, $RO_2$ and $NO_3$ radicals (Emmerson et al., 2007; Ren et al., 2008; Elshorbany et al., 2009; Sommariva et al., 2009; Lu et al., 2012). The ROPA/RODA model output consists of two formatted files with the rate of

formation and loss of a given species for each reaction in which it is present as product or reactant, respectively. The species for which the calculated concentrations and the rate of production/destruction analysis are required are chosen by the user in the model configuration, together with the corresponding output frequency (Sect. 2.5).

    All output files are simple space-delimited text files, which can be easily imported into external data analysis software for further processing and plotting. In AtChem2 the output files are saved in a directory specified by the user when the model run

is started, while in AtChem-online the output files have to be downloaded from the web server as a compressed zip file. Simple plotting tools – in Python, R, MATLAB, gnuplot – allow the user to have a quick look at the model results and at the diagnostic variables as soon as the model run is completed.

## 3   Applications of the AtChem model

### 3.1   Chamber studies

AtChem was originally conceived as a modelling tool for environmental chambers, in order to aid in the characterisation of the chambers, in the interpretation of the experimental results and in the evaluation/development of the MCM (Sect. 1). We demonstrate this type of application using data from a propene oxidation experiment conducted in the Chamber for Experi-





mental Multiphase Atmospheric Simulation (CESAM), at the Laboratoire Inter-universitaire des Systèmes Atmosphériques, near Paris, France (http://www.cesam.cnrs.fr).

The propene chemical mechanism and the inorganic chemistry scheme were extracted from the MCM v3.3.1 and complemented with an auxiliary mechanism specific to the CESAM chamber, as described in Wang et al. (2011). Chamber-specific reactions are needed to model this type of experiment so that the background reactivity of the environmental chamber can be taken into account. This allows the separation of the chamber-specific chemical processes from the underlying processes that are being studied in the experiments, in order to make the results from experiments carried out in different chambers comparable and transferable to the atmosphere. The chamber-specific mechanism for CESAM includes: chamber dilution, loss of $O_3$,

and conversion of $NO_2$ to $NO$ + HONO on the chamber wall, with an initial concentration of HONO of 8 ppbv (Wang et al., 2011). CESAM is an indoor atmospheric simulation chamber and uses three 4 kW Xenon arc lamps as a light source. The photolysis rate of $NO_2$ was the only photolysis rate measured inside the chamber: during the propene oxidation experiment, when the chamber lamps were on, $j(NO_2)$ was a constant value of $3.5 \times 10^{-3}$ s$^{-1}$. The model was constrained to the $j(NO_2)$

measurements, while the remaining photolysis rates were calculated by AtChem using the MCM parametrization and scaled using the `JFAC` correction factor (Sect. 2.4).

Figure 6 shows the modelled mixing ratios of the precursor VOC (propene), with the primary oxidation products HCHO and $CH_3CHO$, the secondary product peroxyacetyl nitrate (PAN, formed via the OH + $CH_3CHO$ reaction), plus the inorganic species NO, $NO_2$ and $O_3$. The propene loss began when the chamber lamps were switched on – 1800 seconds since the

start of the experiment – and was driven by reaction with OH, produced from HONO photolysis. HONO was formed in the chamber from heterogeneous chemistry occurring on the chamber wall; its role in initiating the oxidation of propene demonstrates that it is essential to understand, and include in the model, the chamber-specific chemical mechanism. The model showed good agreement with the observations of propene, NO, $NO_2$ and $CH_3CHO$, with a tendency to overestimate HCHO and underestimate $O_3$ and PAN in the latter stage of the experiment (Fig. 6), which may hint at potential problems with the

chemistry of the oxidation products of propene in the MCM and/or with the chamber auxiliary mechanism. Such experiments can be used to refine and optimise the chamber-specific mechanisms, but, overall, the model results indicate that the MCM is reasonably accurate in its description of the gas-phase oxidation of propene in the troposphere.

### 3.2    Field studies

The chamber experiment and the corresponding model simulation shown in Sect. 3.1 are relatively simple: the chemical mech-

anism only had 83 species and 261 reactions, the model was unconstrained (except for $j(NO_2)$) and the duration of the experiment was less than 2 hours. Intensive field campaigns typically last for several days or weeks and the chemical mechanism needed for a campaign model is usually much larger than the one needed for a chamber model. It is not unusual for a campaign model to use the entire MCM (>17000 chemical reactions), along with a hundred or more constrained variables. This makes the model computationally very expensive and difficult to run on a web application, such as AtChem-online.

AtChem2 was developed specifically for the long and complex simulations needed for field studies. We demonstrate this type of application using the dataset of the Texas Air Quality Study 2006, an intensive ship-based field campaign on the Gulf





Coast of the United States (Parrish et al., 2009). The cruise took place between 27 July and 11 September 2006 on the NOAA research vessel *Ronald H. Brown*; the radical measurements (total peroxy radicals and $NO_3$) and the corresponding modelling work are discussed in Sommariva et al. (2011a).

The chemical mechanism was extracted from the MCM v3.1 and included the inorganic chemistry scheme, the oxidation mechanism of 65 VOC, plus the dimethyl sulphide (DMS) oxidation mechanism from Sommariva et al. (2009). In addition, dry deposition and heterogeneous reactions for the appropriate gas-phase species were included in the chemical mechanism. The model configuration and constraints – chemical species, photolysis rates, aerosol surface area, temperature, water vapour, boundary layer height – were the same as in the model described by Sommariva et al. (2011a). In that work, the model showed

reasonably good agreement with the measured concentrations of total peroxy radicals (within ∼30%, on average), although it underestimated the measurements of $NO_3$ by approximately a factor of 3.

The modelled concentrations of total peroxy radicals ($HO_2+RO_2$) for the period July 31-August 2 are shown in Fig. 7, together with the corresponding measurements. The results of AtChem2 and of the version of AtChem used by Sommariva et al. (2011a) differ by ∼3% – a discrepancy due to a small bug in the calculation of `JFAC` which was fixed in a later version of

AtChem. During the day, the model overestimated the measured concentrations of $HO_2+RO_2$ by 10-25%, which is well within the instrumental uncertainty (∼40%). During the night, the model underestimated the measurements of $HO_2+RO_2$ by up to 57%, although the disagreement between the model and the measurements at nighttime during the entire cruise was on average lower (25-30%, Sommariva et al. (2011a)). The ability of the model to reproduce the observations of total peroxy radicals provides useful insight into the chemical processes in the marine boundary layer: the model-measurements discrepancies

indicate that, under unpolluted conditions, radical chemistry is much better understood during the day than during the night, which suggests that future studies should focus on nocturnal chemistry.

The ROPA/RODA model output (Sect. 2.6) can be used to investigate the details of the chemical processes in the unpolluted marine atmosphere encountered during the first few days of TexAQS 2006. The model results indicate that, in that period, the methyl peroxy radical ($CH_3O_2$) was the major component of the $RO_2$ pool (30-45% during the day, 50-80% during the night).

Figure 8 shows the rates of production and destruction of $CH_3O_2$ at midday and midnight of August 1, when the ship was in the Atlantic Ocean off the coast of Florida. The main destruction term for $CH_3O_2$ was the reaction with NO, even though the levels of nitrogen oxides were low during the first few days of the cruise ($<1$ ppbv, on average). The reactions of $CH_3O_2$ with $HO_2$ and $RO_2$ accounted together for about half of the total $CH_3O_2$ loss at midday, but for only ∼15% at midnight, because of the very low nocturnal concentrations of peroxy radicals. It must be noted, however, that since the model underestimated the

concentrations of $HO_2$ and $RO_2$ during the night (Fig. 7), their role as $CH_3O_2$ sinks was also underestimated.

The oxidation of methane and the reactions of the acetyl peroxy radical – $CH_3CO_3$, typically formed from the oxidation of C2-C5 hydrocarbons – with NO and other peroxy radicals were the major sources of $CH_3O_2$. During the day, the oxidation of carbonyls and of organic acids was a significant contributor to the formation of $CH_3O_2$; at night, methane oxidation was driven by OH radicals formed by the ozonolysis of alkenes, while DMS oxidation (mostly via reaction with $NO_3$, Sommariva et al.

(2011a)) accounted for up to a third of the total $CH_3O_2$ production. The formation pathways of the methyl peroxy radical in the unpolluted marine atmosphere highlight the different chemical processes taking place during the day, when OH photochemistry



dominates, and during the night, when reactions initiated by $NO_3$ and $O_3$ become an important source of short-chain organic peroxy radicals.

## 4   Summary and future work

AtChem provides a tool to model atmospheric chemical processes that is free, open source, quick to set up and easy to use. Semi-automated scripts and simple text files allow the user to install, configure and run an atmospheric chemistry box-model even with little modelling experience. A particular strength of AtChem is the ease with which models can be constrained to measured data and the possibility to use constraints with different timescales, a feature that allows the user to exploit all the information contained in the measurements and greatly decreases the time needed to prepare and pre-process the model

constraints. Another important component of AtChem is the implementation of a continuous integration workflow, which – together with a comprehensive suite of tests and version control software – allows the model results to be verified against known solutions, as well as to track and record all the modifications to the code. This ensures that changes to the AtChem codebase are fully documented and do not cause unintended behaviour, thus making AtChem robust, reliable and traceable. Although primarily designed for the MCM, AtChem can be as easily adapted to use any other chemical mechanism, as long as

it is provided in the correct format.

There are two versions of AtChem available: AtChem-online runs as a web application (https://atchem.leeds.ac.uk/webapp/) and is suitable for relatively simple simulations, such as laboratory and environmental chamber experiments. AtChem2 is a development of AtChem-online designed to run more complex and longer simulations, such as ambient measurements and field campaigns, and to facilitate batch simulations for sensitivity studies. AtChem2 is available at https://github.com/AtChem/,

under the open source MIT license. We demonstrate the capabilities of AtChem to model chamber experiments and field studies with examples taken from the EUROCHAMP database and the NOAA TexAQS 2006 field campaign, respectively.

Future work and development plans for AtChem2 include:

–   Implementation of a system to read the chemical mechanism at runtime, which will eliminate the need to recompile the executable more than once (unless the underlying Fortran source code is modified) and further simplify batch model

runs.

–   Expansion of the test suite and detailed profiling of the code at runtime to identify and streamline bottlenecks and make the model faster to run.

–   Simplification of the model configuration and output, and addition of different formats for the chemical mechanism, such as the format used by the open source modelling software KPP (Damian et al., 2002).

In addition, AtChem-online needs to be upgraded to the AtChem2 codebase with a new and improved Web Interface. A more simple version of the upgraded AtChem-online may also be developed for educational and outreach purposes: this version should feature a basic user interface, simplified configuration options and more intuitive visualisation tools.



*Code and data availability.* The AtChem-online code and documentation are available at https://atchem.leeds.ac.uk/webapp/. The AtChem2 code and documentation are available at https://github.com/AtChem/. This work contains data from the EUROCHAMP Database of Atmo-
spheric Simulation Chamber Studies (DASCS, https://data.eurochamp.org/) at CNRS-AERIS and the NOAA-ESRL Tropospheric Chemistry Measurements Database (https://esrl.noaa.gov/csd/groups/csd7/measurements/).

*Author contributions.* CM, KB, JY, PJ, MJP and ARR designed and developed AtChem-online. RS and SC developed AtChem2 from AtChem-online. VNM, MJN and MP tested the AtChem code and ran the simulations. RS, ARR, WJB and PSM prepared the manuscript with substantial constributions from the other authors.

*Competing interests.* The authors declare no conflicts of interest.

*Acknowledgements.* We thank P. Bräuer, B. Nelson (University of York), M. Vázquez-Moreno (CEAM/EUPHORE), D. Waller and R. Woodward-Massey (University of Leeds) for their contributions and feedback. We also thank J. Wakelin and the University of Leicester Research Software Engineering Team (ReSET) for their support. Many thanks to H. Stark (University of Colorado-Boulder, USA) and J.-F. Doussin (Université Paris-Est Créteil, France) for the datasets used to test and demonstrate the model. ARR and MJN acknowledge funding
from the EU Horizon 2020 research and innovation programme through the EUROCHAMP-2020 Infrastructure Activity (grant agreement No. 730997).





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





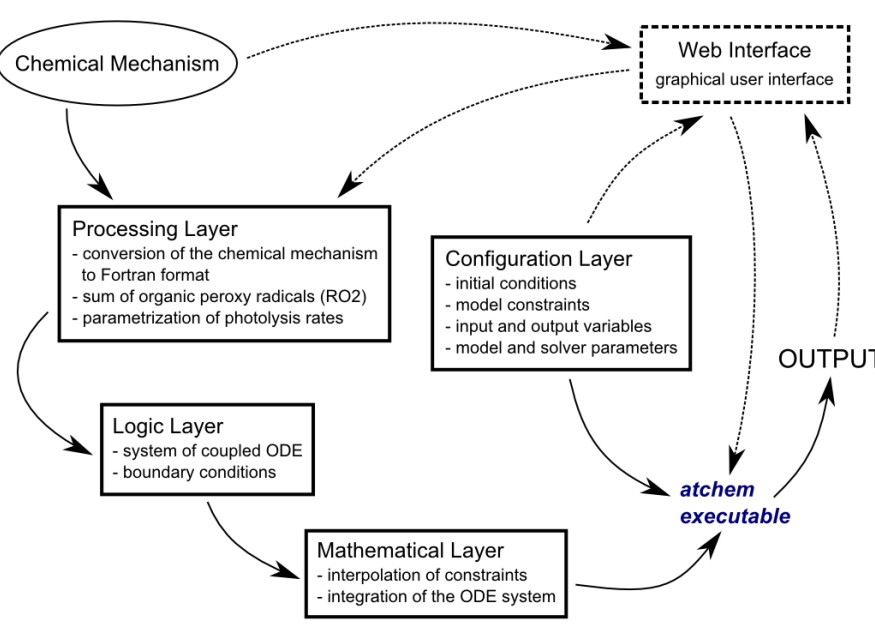

**Figure 1.** Structure of the AtChem model. The dashed lines indicate the model components that are present in AtChem-online, but not in AtChem2.



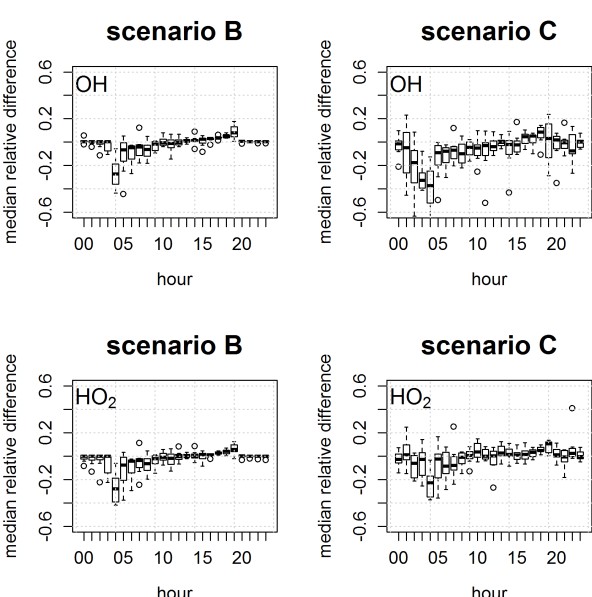

**Figure 2.** Diurnal distributions of the relative differences in the calculated concentrations of OH and $HO_2$ in scenario B and C compared to scenario A (Table 1) over a 9 days model run. The box-and-whiskers show the medians, and the 1st and 3rd quartiles, while the open circles indicate the outliers.





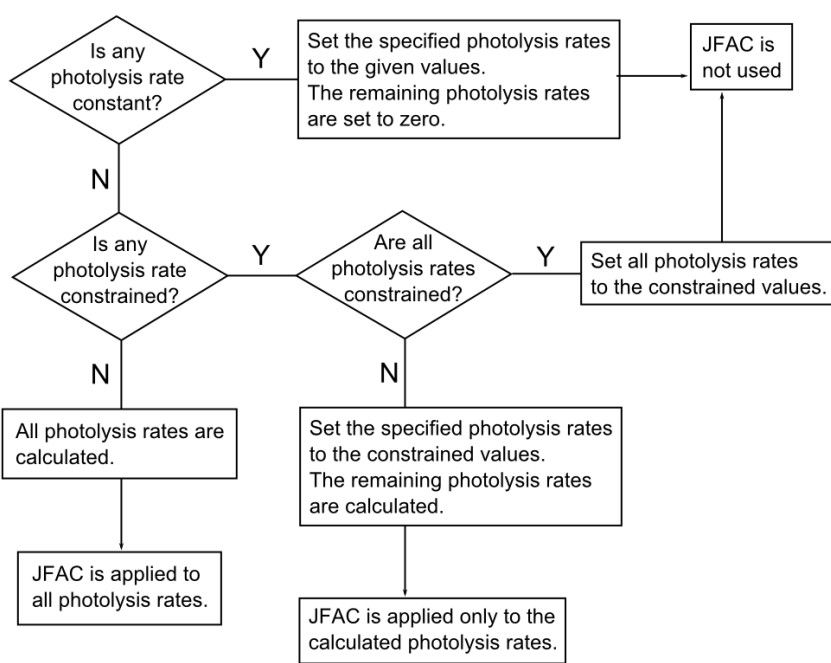

**Figure 3.** Treatment of photolysis rates in AtChem.



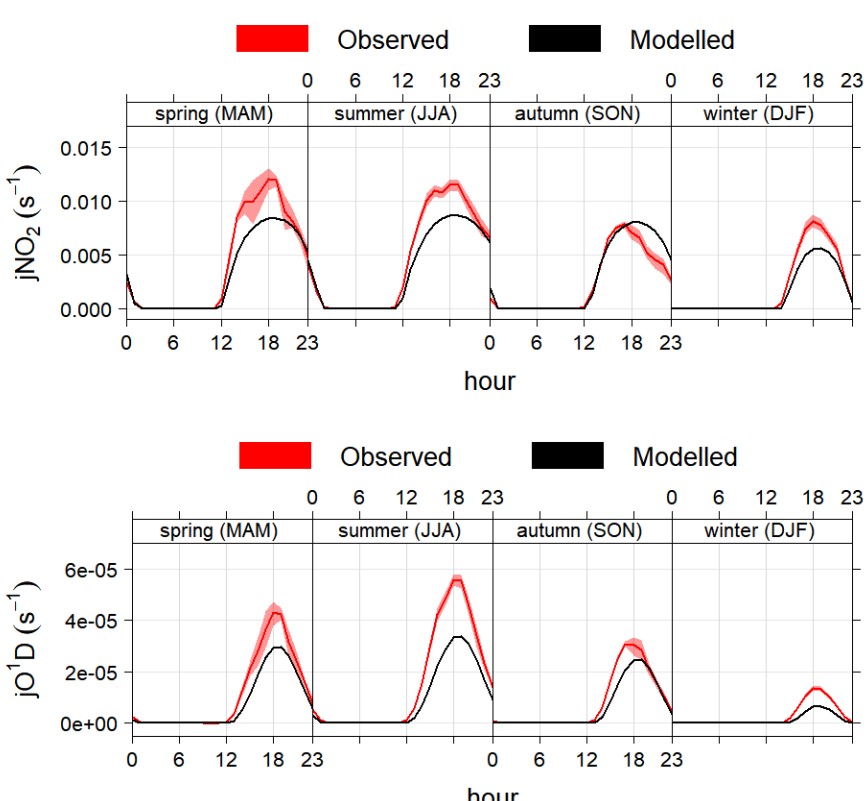

**Figure 4.** Average modelled and measured $j(NO_2)$ and $j(O^1D)$ during different seasons in Boulder, CO (USA). The shaded areas are the 95% confidence intervals of the mean. The timestamp is in Greenwich Mean Time, which is the timezone used by AtChem (local time is GMT-7).





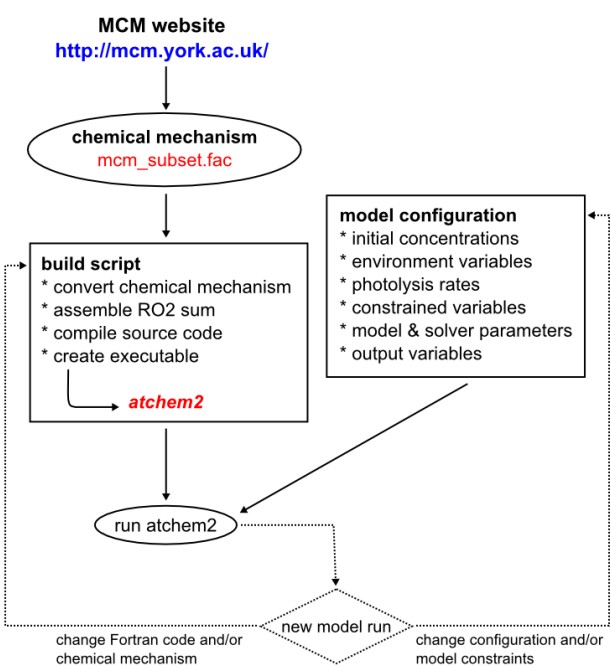

**Figure 5.** Workflow of the AtChem2 model.



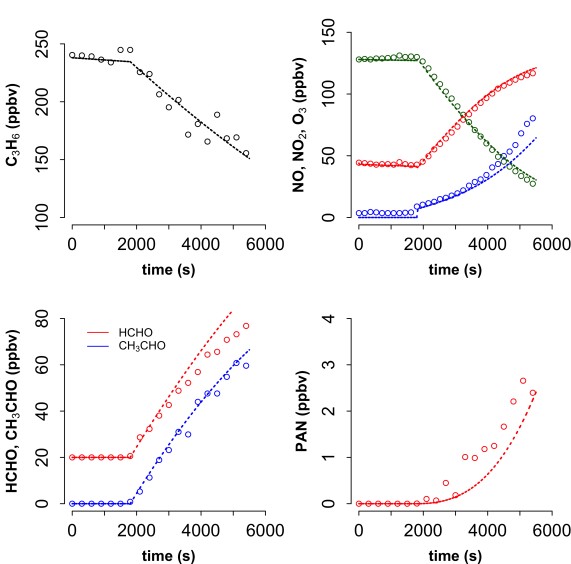

**Figure 6.** Measured (points) and modelled (lines) mixing ratios of propene ($C_3H_6$), ozone ($O_3$), nitrogen oxides (NO, $NO_2$) and propene oxidation products (HCHO, $CH_3CHO$, PAN) during a propene oxidation experiment at the CESAM atmospheric simulation chamber.





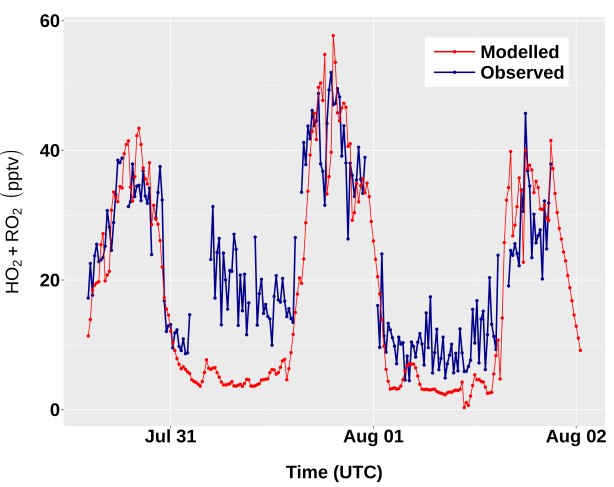

**Figure 7.** Measured and modelled concentrations of total peroxy radicals ($HO_2$+$RO_2$) between July 31 and August 2, during the TexAQS 2006 cruise of the NOAA research vessel *Ronald H. Brown*.





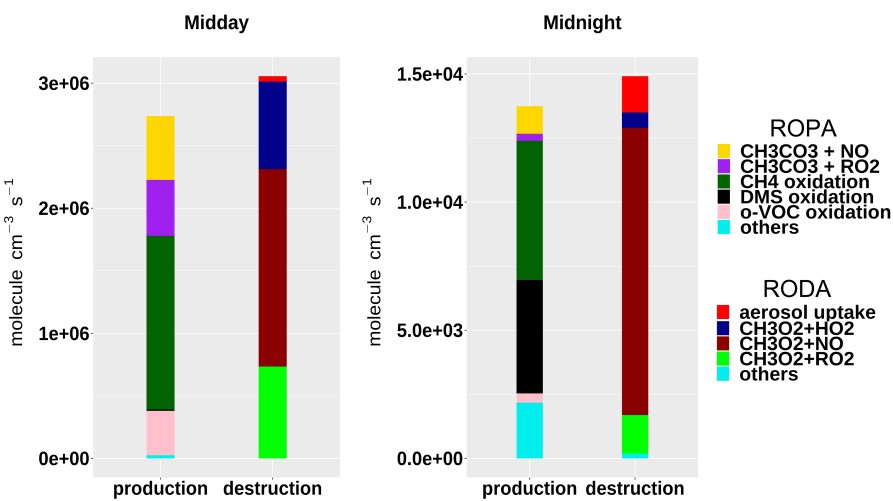

**Figure 8.** Rate of production (ROPA) and destruction (RODA) analysis of the methyl peroxy radical (CH$_3$O$_2$) at midday and midnight of August 1, during the TexAQS 2006 cruise of the NOAA research vessel *Ronald H. Brown*.





**Table 1.** Frequencies of the original measurements and averaged frequencies of the constrained data used in each model scenario.

| Constraint | Measurements Frequency (min) | Constraints Frequency (min) | | |
|---|---|---|---|---|
| | | Scenario A | Scenario B | Scenario C |
| Photolysis Rates | 1 | 15 | 1 | 1 |
| Environment Variables § | 1 | 15 | 15 | 1 |
| $O_3$, NO, $NO_2$, SO2 | 1 | 15 | 15 | 1 |
| CO, $H_2$ | 5 | 15 | 15 | 15 |
| $CH_4$ | 20 | 60 | 60 | 60 |
| VOC (PTR-MS) † | 2 | 15 | 15 | 15 |
| VOC (GC-MS) ‡ | 60 | 60 | 60 | 60 |

§ temperature, pressure, relative humidity, sun declination.

† C1-C4 oxygenated hydrocarbons.

‡ C2-C7 hydrocarbons.