# Peer review of "AtChem (version 1), an open source box-model for the Master Chemical Mechanism"

_Geoscientific Model Development, 2019_

## Short Comment (SC1) · 7 Aug 2019

Dear authors,

in my role as Executive editor of GMD, I would like to bring to your attention our Editorial version 1.2:

https://www.geosci-model-dev.net/12/2215/2019/

This highlights some requirements of papers published in GMD, which is also available on the GMD website in the 'Manuscript Types' section: http://www.geoscientific-model-development.net/submission/manuscript_types.html

In particular, please note that for your paper, the following requirement has not been

met in the Discussions paper:

- "The main paper must give the model name and version number (or other unique identifier) in the title."

Please add the AtChem version number to the title of your revised manuscript

Additionally, please note, that GMD is encouraging authors to provide a persistent access to the exact version of the source code used for the model version presented in the paper. As explained in https://www.geoscientific-model-development.net/about/manuscript_types.html the preferred reference to this release is through the use of a DOI which then can be cited in the paper. For projects in GitHub a DOI for a released code version can easily be created using Zenodo, see https://guides.github.com/activities/citable-code/ for details.
Finally note, that according to our new Editorial (v1.2) all data and analysis / plotting scripts should be made available.

Yours, Astrid Kerkweg

―――――――――――――――――

---

## Referee Comment (RC1) · Anonymous Referee #1 · 10 Oct 2019

This paper documents a boxmodeling system for the Master Chemical Mechanism (MCM). The MCM is a very large, near explicit mechanism describing the degradation of 143 primary compounds into $CO_2$ and $H_2O$. As such, the MCM is too computationally expensive to be run in global chemistry models or even air quality models. However the MCM is often used to benchmark smaller 'lumped' chemistry schemes by comparison using a zero-dimensional boxmodel, or by comparison with results from controlled chamber experiments. Much of the boxmodeling work in the past used expensive commercial software, and this new open source system was developed in part to avoid these costs, but also to provide potential additional capability beyond commercial 'black-box' systems.

The paper is generally well written, with examples provided on how the model works

and performs. However, AtChem is presented as being a new and novel free system (line 52) without acknowledging any of the recent efforts to produce similar open source boxmodels, some if not all are also designed to run the MCM. These models are missing from the literature review. The literature cited between lines 23-29 are studies using boxmodels, some were commercial like FACSIMILE, some not. The review does not cover the models themselves. I have scored 'fair' for scientific significance (substantial new concepts, ideas, or methods) and scientific quality (consideration of related work, including appropriate references) based on this and the decision not to upgrade the representation of photolysis, which many of the other open source boxmodels have done (see major questions).

If the paper is to be published, acknowledgement of previous work towards production of open source boxmodels should be included in the literature review. I also would suggest the authors look into (at least comparing) other photolysis schemes.

Major questions

More than 10 years ago, the MCM website moved to output the entire mechanism (or a chosen subset) in a variety of formats. One of those formats was KPP (kinetic PreProcessor software) which is the basis for some of these recent open source codes – Knote et al's (2015) boxmodel extensions to KPP (BOXMOX, https://www2.acom.ucar.edu/modeling/boxmox-box-model-extensions-kpp), Sander et al's (2011) Chemistry As A Box Model Application (CAABA), and the Dynamically Simple Model of Atmospheric Chemical Complexity (DSMACC, http://wiki.seas.harvard.edu/geos-chem/index.php/DSMACC_chemical_box_model) (Emmerson and Evans, 2009). There is a wiki page listing 10 of these models: https://en.wikipedia.org/wiki/Kinetic_PreProcessor I am also aware of boxmodels which do not use these KPP format codes – F0AM (Wolfe et al., 2016) uses Matlab. So I'm missing why the authors developed their own model from scratch rather than building on another system which is freely available?

The method for calculating photolysis rates in AtChem is a shortcoming. AtChem uses (quite an old) 2-stream method for calculating photolysis rates, and looking at figure 4, consistently underpredicts the measured rates. This photolysis scheme has been used with MCM modeling for ~20 years or so. Yes, it is important to adjust for cloudy conditions using measured photolysis rates. However not all investigators (and particularly students using AtChem in the classroom) have the luxury of being able to measure photolysis rates to perform the correction in equation 4, or to use directly as a constraint. If equation 3 can only be relied upon to produce good results at 500 m altitude, 45 degrees N on July 1st, then why was the photolysis method not updated given the opportunity for designing this new system? If AtChem had used one of the other open source systems as a basis, photolysis could be calculated on-line, where such parameters can be changed. CAABA gives users a range of options for calculating photolysis - JVAL, RADJIMT, DISSOC, (Sander et al., 2019). There's also FAST-JX (Wild et al., 2000) which users of GEOSChem and the UK chemistry and aerosol (UKCA) community prefer (https://www.ess.uci.edu/researchgrp/prather/scholar_software/fast-jx). BOXMOX and DSMACC use the Tropospheric Ultraviolet and Visible radiation (TUV, (Madronich and Flocke, 1997)), now at version 5.3.2 https://www2.acom.ucar.edu/modeling/tuv-download. i.e., there are plenty of other systems available. Given that photolysis is so important to OH production, correcting the underestimation produced by the 2-steam method is crucial for AtChem to be of use to other researchers, and should be considered.

As this is a model description paper, please explain how the dilution factor is calculated. In a number of places in the text constraining the boundary layer height is mentioned which would impact the chemical concentrations, but it is not explained. This also applies to the chamber open roof experiments. Please also mention whether the model has a capability for entrainment of stratospheric air into the troposphere (which is a feature of BOXMOX), or exchange of air with air masses outside the box (a feature of CAABA).

The choice of examples shown to demonstrate the AtChem system are at odds with the description of why it was developed. For example line 345 "AtChem2 was developed specifically for the long and complex simulations need for field studies". AtChem2 is then demonstrated using only 2 days from a ∼40 day TEXAQs campaign. Were these two days chosen because they provided the best model-observation comparison? Why not show the whole TEXAQs time series, which would show that AtChem has been rigorously tested?

Editorial comments:

Title. I think GMD prefers a version number to be assigned to the model being described.

line 174 represents the start of a new paragraph but mentions 'this' modeling technique, which is not defined, or must relate to the paragraph above. I assume the latter, in which case it isn't a new paragraph.

Line 204. 'study' not studies

Line 268. "since the model. . ..' This and the following sentence are both very long, and could be broken up.

Line 350. Why is the latest version of the MCM not being used here?

Line 371. The reader needs to know what compounds were being constrained here – for example I'm assuming NO was constrained because of the statement about NO being the main destruction term. Earlier in this section reference is given to the model set-up in Sommariva et al (Sommariva et al., 2011), but as the results here rely upon the constraints they should be stated.

Figure 2. the plots are too small to be seen properly.

Figure 6, top right panel. A legend is missing for the three colors.

References

Emmerson, K. M. and Evans, M. J.: Comparison of tropospheric gas-phase chemistry schemes for use within global models, Atmospheric Chemistry and Physics, 9(5), 1831–1845, doi:https://doi.org/10.5194/acp-9-1831-2009, 2009.

Knote, C., Tuccella, P., Curci, G., Emmons, L., Orlando, J. J., Madronich, S., Baró, R., Jiménez-Guerrero, P., Luecken, D., Hogrefe, C., Forkel, R., Werhahn, J., Hirtl, M., Pérez, J. L., San José, R., Giordano, L., Brunner, D., Yahya, K. and Zhang, Y.: Influence of the choice of gas-phase mechanism on predictions of key gaseous pollutants during the AQMEII phase-2 intercomparison, Atmospheric Environment, 115, 553–568, doi:10.1016/j.atmosenv.2014.11.066, 2015.

Madronich, S. and Flocke, S.: Theoretical Estimation of Biologically Effective UV Radiation at the Earth's Surface, in Solar Ultraviolet Radiation, edited by C. S. Zerefos and A. F. Bais, pp. 23–48, Springer Berlin Heidelberg., 1997.

Sander, R., Baumgaertner, A., Gromov, S., Harder, H., Jöckel, P., Kerkweg, A., Kubistin, D., Regelin, E., Riede, H., Sandu, A., Taraborrelli, D., Tost, H. and Xie, Z.-Q.: The atmospheric chemistry box model CAABA/MECCA-3.0, Geoscientific Model Development, 4(2), 373–380, doi:https://doi.org/10.5194/gmd-4-373-2011, 2011.

Sander, R., Baumgaertner, A., Cabrera-Perez, D., Frank, F., Gromov, S., Grooß, J.-U., Harder, H., Huijnen, V., Jöckel, P., Karydis, V. A., Niemeyer, K. E., Pozzer, A., Riede, H., Schultz, M. G., Taraborrelli, D. and Tauer, S.: The community atmospheric chemistry box model CAABA/MECCA-4.0, Geosci. Model Dev., 12(4), 1365–1385, doi:10.5194/gmd-12-1365-2019, 2019.

Sommariva, R., Bates, T. S., Bon, D., Brookes, D. M., de Gouw, J. A., Gilman, J. B., Herndon, S. C., Kuster, W. C., Lerner, B. M., Monks, P. S., Osthoff, H. D., Parker, A. E., Roberts, J. M., Tucker, S. C., Warneke, C., Williams, E. J., Zahniser, M. S. and Brown, S. S.: Modelled and measured concentrations of peroxy radicals and nitrate radical in the U.S. Gulf Coast region during TexAQS 2006, J Atmos Chem, 68(4), 331–362, doi:10.1007/s10874-012-9224-7, 2011.

Wild, O., Zhu, X. and Prather, M. J.: Fast-J: Accurate Simulation of In- and Below-Cloud Photolysis in Tropospheric Chemical Models, Journal of Atmospheric Chemistry, 37(3), 245–282, doi:10.1023/A:1006415919030, 2000.

Wolfe, G. M., Marvin, M. R., Roberts, S. J., Travis, K. R. and Liao, J.: The Framework for 0-D Atmospheric Modeling (F0AM) v3.1, Geosci. Model Dev., 9(9), 3309–3319, doi:10.5194/gmd-9-3309-2016, 2016.

---

## Referee Comment (RC2) · Anonymous Referee #2 · 12 Oct 2019

**1 General comments**

The paper describes a new open-source box model designed primarily for runs using the Master Chemical Mechanism (MCM), although other mechanisms can also be modelled provided they are entered in the correct format. The model comes in two forms, one (the original) available online via a web-interface, and a downloadable one (AtChem2) suitable for more advanced studies including batch runs that can be executed on the user's machine. The box model is presented as being free and easy to use, and having functionality that makes it particularly suitable for comparison against observational data, due to the way it handles constrained variables with different time frequencies. It is applied in two case studies to chamber and field data respectively to

illustrate that it can handle these scenarios and allow us to reach some conclusions about which parts of the mechanism need to be reviewed.

I believe this paper will be suitable for publication subject to revision.

**2 Specific comments**

The text in the paper is generally well-written and easy to follow. The figures are less clear, however. In particular, Figure 2 is too small to make much sense of even when zoomed in on (due to the scale); the outlier marker is larger than some of the box-and-whiskers. Figure 3 is missing some arrowheads, and the ones that are there are too small to be clearly seen. In Figure 6, the ozone and nitrogen oxides lines/markers are not labelled as to which is which; I would also recommend using different marker shapes when presenting multiple datasets in one graph in addition to using different colours.

I think the paper could highlight in detail what makes AtChem stand out over specific other box models. The two forms (online/offline) appear to have different unique benefits, and perhaps need to be discussed separately. It appears that the online model can at present only be run with login details (is it only for use by the EUROCHAMP community?), which might restrict takeup (especially for teaching purposes), but I hope this is something the authors are planning to address. I also note that the online form does not seem to readily support the newest MCM version; perhaps this can be updated?

Parts of the paper are using different versions of the MCM and perhaps also different versions of AtChem(2). In particular, on page 12 the section on lines 358-360 describes that the previously published results were off by a small amount due to a bug in a previous version of AtChem; however, at no point does the paper make reference to which version(s) of AtChem(2) were used in the runs. I would suggest carrying out all the model runs with the latest version of AtChem2 and annotating them as such. If

the latest MCM version was not used for modelling the field study data for consistency with the previous paper analysing this data, I would comment on this, but also perhaps rerun the simulation with the newest MCM version for comparison.

**3   Technical corrections**

p7, line 204: change "studies" to "study"

p7, line 214: change "results" to "result" and correct spelling of "stiffness"

p8, line 224: Eq. 3 appears to use two different forms of the multiplication sign

p12, line 351: IUPAC spelling is "sulfide"

p22, label on Figure 4: the local time is GMT–7 in the winter, but GMT–6 in the summer due to daylight saving time

p27, in Table 1: "SO2" should have 2 as a subscript

There is inconsistent use of American and British spellings (s/z) in the manuscript, and though it is a very minor issue, this can perhaps be standardised. I also feel that a few more hyphens would aid easy parsing of the text in places (e.g. p7, line 205 as well as Figure 2 label: "9-days"; p9, line 250 as well as p8, line 226: "two-stream").

---

## Author Comment (AC1) · 19 Nov 2019

Dear authors, in my role as Executive editor of GMD, I would like to bring to your attention our Editorial version 1.2: https://www.geosci-model-dev.net/12/2215/2019/.

This highlights some requirements of papers published in GMD, which is also available on the GMD website in the 'Manuscript Types' section: http://www.geoscientific-model-development.net/submission/manuscript_types.html. In particular, please note that for your paper, the following requirement has not been met in the Discussions paper: "The main paper must give the model name and version number (or other unique identifier) in the title." Please add the AtChem version number to the title of your revised manuscript

We thank the editor for bringing this point to our attention. The title of the manuscript has been changed to: "AtChem (version 1), an open source box-model for the Master Chemical Mechanism".

Please also note that Beth S. Nelson has been added as co-author and more information on the version of AtChem presented in the manuscript has been added to the text, following comments by both referees.

Additionally, please note, that GMD is encouraging authors to provide a persistent access to the exact version of the source code used for the model version presented in the paper. As explained in https://www.geoscientific-model-development.net/about/manuscript_types.html the preferred reference to this release is through the use of a DOI which then can be cited in the paper. For projects in GitHub a DOI for a released code version can easily be created using Zenodo, see https://guides.github.com/activities/citable-code/ for details.

A DOI (10.5281/zenodo.3404021) has been created on Zenodo and added to the GitHub release page and to Section 2 of the manuscript.

Finally note, that according to our new Editorial (v1.2) all data and analysis / plotting scripts should be made available.

All data used in the manuscript are available at the repositories indicated in the "Code and data availability." section. Plotting scripts are not relevant in this case.

---

## Author Comment (AC2) · 19 Nov 2019

This paper documents a boxmodeling system for the Master Chemical Mechanism (MCM). The MCM is a very large, near explicit mechanism describing the degradation of 143 primary compounds into $CO_2$ and $H_2O$. As such, the MCM is too computationally expensive to be run in global chemistry models or even air quality models. However the MCM is often used to benchmark smaller 'lumped' chemistry schemes by comparison using a zero-dimensional boxmodel, or by comparison with results from controlled chamber experiments. Much of the boxmodeling work in the past used expensive commercial software, and this new open source system was developed in part to avoid these costs, but also to provide potential additional capability beyond commercial 'black-box' systems. The paper is generally well written, with examples provided on how the model works and performs. However, AtChem is presented as being a new and novel free system (line 52) without acknowledging any of the recent efforts to produce similar open source boxmodels, some if not all are also designed to run the MCM. These models are missing from the literature review. The literature cited between lines 23-29 are studies using boxmodels, some were commercial like FACSIMILE, some not. The review does not cover the models themselves. I have scored 'fair' for scientific significance (substantial new concepts, ideas, or methods) and scientific quality (consideration of related work, including appropriate references) based on this and the decision not to upgrade the representation of photolysis, which many of the other open source boxmodels have done (see major questions). If the paper is to be published, acknowledgement of previous work towards production of open source boxmodels should be included in the literature review. I also would suggest the authors look into (at least comparing) other photolysis schemes.

We thank the referee for their comments and suggestions. Please find below our replies and the related modifications to the manuscript. The line numbers refer to the version of the manuscript published on GMDD.

**Major questions**

More than 10 years ago, the MCM website moved to output the entire mechanism (or a chosen subset) in a variety of formats. One of those formats was KPP (kinetic PreProcessor software) which is the basis for some of these recent open source codes – Knote et al's (2015) boxmodel extensions to KPP (BOXMOX, https://www2.acom.ucar.edu/modeling/boxmox-box-model-extensions-kpp), Sander et al's (2011) Chemistry As A Box Model Application (CAABA), and the Dynamically Simple Model of Atmospheric Chemical Complexity (DSMACC, http://wiki.seas.harvard.edu/geos-chem/index.php/DSMACC_chemical_box_model)
(Emmerson and Evans, 2009). There is a wiki page listing 10 of these models: https://en.wikipedia.org/wiki/Kinetic_PreProcessor I am also aware of boxmodels which do not use these KPP format codes – F0AM (Wolfe et al., 2016) uses Matlab. So I'm missing why the authors developed their own model from scratch rather than building on another system which is freely available?

References to other open source modelling frameworks have been added to the manuscript. With regard to the question of why AtChem was developed from scratch instead of building on other work, it should be noted that AtChem-online has been available for some time. The first version of the model was released in 2010, although there was not an accompanying publication at the time.

Although this has changed in the past few years (as the referee points out), at the time the choice of modelling tools was rather limited and the available ones were expensive and/or not very easy to use for novice modellers. As explained in the introduction (lines 50-60) and at the beginning of Section 2 (lines 75-76), AtChem-online was created as a free community modelling toolkit to facilitate the use of the MCM within the EUROCHAMP atmospheric

chamber community, by providing a tool that could be used by researchers without an experienced modelling background.

The offline version, Atchem2, was released more recently (in 2017) and is based on the same code as AtChem-online. Therefore, Atchem2 is not a new model, but a development and an improvement of a pre-existing model. Given the fact that a paper about AtChem-online had not been published, it just makes sense that the older online version of the model is presented together with the newer offline version. This has the added benefit of documenting the model developments within the formal literature.

The final paragraph of the Introduction has been modified as follow:

"AtChem was conceived with the above principles and objectives in mind: the code is free, open source and publicly available. It was released online in 2010, introduced to the EUROCHAMP community via a two day workshop, and briefly described in the annual EUROCHAMP report in late 2010. In recent years, a number of other open source modelling tools and frameworks have been released: some include their own chemical mechanism (e.g., CAABA, Sander et al. (2011)), while others are designed to use primarily the MCM (e.g., PyBox, Topping et al. (2018)). Most these tools – such as DSMACC (Emmerson and Evans, 2009), BOXMOX (Knote et al., 2015), and F0AM (Wolfe et al., 2016) – give the user the flexibility to run different chemical mechanisms. Although AtChem was designed mainly to encourage the use of the MCM in atmospheric chemistry studies (and hence to facilitate its evaluation by the community), it can be easily adapted to model other chemical systems and to use other chemical mechanisms, as long as they are provided in the correct format."

The method for calculating photolysis rates in AtChem is a shortcoming. AtChem uses (quite an old) 2-stream method for calculating photolysis rates, and looking at figure 4, consistently underpredicts the measured rates. This photolysis scheme has been used with MCM modeling for ~20 years or so. Yes, it is important to adjust for cloudy conditions using measured photolysis rates. However not all investigators (and particularly students using AtChem in the classroom) have the luxury of being able to measure photolysis rates to perform the correction in equation 4, or to use directly as a constraint. If equation 3 can only be relied upon to produce good results at 500 m altitude, 45 degrees N on July 1st, then why was the photolysis method not updated given the opportunity for designing this new system? If AtChem had used one of the other open source systems as a basis, photolysis could be calculated on-line, where such parameters can be changed. CAABA gives users a range of options for calculating photolysis – JVAL, RADJIMT, DISSOC, (Sander et al., 2019). There's also FAST-JX (Wild et al., 2000) which users of GEOSChem and the UK chemistry and aerosol (UKCA) community prefer (https://www.ess.uci.edu/researchgrp/prather/scholar_software/fast-jx). BOXMOX and DSMACC use the Tropospheric Ultraviolet and Visible radiation (TUV, (Madronich and Flocke, 1997)), now at version 5.3.2 https://www2.acom.ucar.edu/modeling/tuv-download. i.e., there are plenty of other systems available. Given that photolysis is so important to OH production, correcting the underestimation produced by the 2-steam method is crucial for AtChem to be of use to other researchers, and should be considered.

There is a misunderstanding here about the calculation of the photolysis rates.

First, it is important to distinguish between what is in the MCM and what is in AtChem. The MCM uses a 2-stream scattering model to calculate the photolysis rates of the relevant species; a fitting procedure to the 2-stream scattering model results is then used to derive the empirical parameters l, m, n, which are used in Equation 3 to calculate the photolysis rates as a function of the solar zenith angle in a computationally efficient manner. This methodology is explained in the MCM protocol papers (Jenkin et al, 1997, Saunders et al., 2003). AtChem does not use the 2-stream scattering model, but simply implements Equation 3.

Second, the values of the empirical parameters l, m, n are not prescribed in AtChem, which takes the values of the version of the MCM that is being used. The reason for this approach is that AtChem is designed to use the MCM as is. We think it is not up to the AtChem developers to correct or fix the MCM; this is a task better left to the MCM developers and documented within the MCM protocol. Moreover, nothing prevents the users from providing their own set of empirical parameters based on their preferred photolysis scheme (as stated on lines 229-230). Alternatively, the photolysis rates can be constrained to measurements or to offline-calculated values, as is frequently done by MCM modellers (see line 232). It is left to the users to decide how they wish to parametrise/constrain the photolysis rates, based on their specific needs.

Finally, we do agree with the referee that the 2-stream scattering model is no longer up to date. Indeed, the next version of the MCM will use an updated version of the TUV model to calculate the photolysis rates. But this does not mean that equation 3 will no longer be used. It simply means that the output of TUV will be used to derive a new set of l, m, n parameters. We have no evidence that equation 3 is not reliable and therefore we see no reason to design a new method to calculate the photolysis rates in AtChem. It is however true that the set of l, m, n used in the current version of the MCM (3.3.1) would benefit from an update.

To make this point clearer the following modifications have been made to the manuscript:

Lines 225-227: sentence changed to "The empirical parameters l, m, n are calculated, for each version of the MCM, as explained by Jenkin et al. (1997) and Saunders et al. (2003): in the MCM v3.3.1 (and previous versions), the empirical parameters are obtained by fitting Eq. 3 to the output of a two-stream isotropic scattering model, which incorporates the appropriate photolysis cross-sections and quantum yields."

Line 230: "to replace the default values of l, m, n and τ ." changed to "to replace the values of l, m, n and τ provided by the MCM"

Line 232: changed to "The photolysis rates can also be set to constant values, constrained to measured data or constrained to values calculated offline using a suitable radiative transfer model"

Line 250: "in Eq. 3" changed to "in the MCM"

As this is a model description paper, please explain how the dilution factor is calculated. In a number of places in the text constraining the boundary layer height is mentioned which would impact the chemical concentrations, but it is not explained. This also applies to the chamber open roof experiments. Please also mention whether the model has a capability for entrainment of stratospheric air into the troposphere (which is a feature of BOXMOX), or exchange of air with air masses outside the box (a feature of CAABA).

A complete explanation of each environment variable can be found in the AtChem manual, which is included with the AtChem code. To clarify the (optional) usage of the variables used to parametrise the emission, dilution and deposition of chemical species, the following paragraph has been added to Section 2.2:

"Chemical reactions can also be written without reactants or products, which is useful to parametrise non-chemical processes in the model, if required. For example, emission of species P1 can be parametrised as:

% Er : = P1 ;

where Er is the emission rate in s-1 . Likewise, dry deposition and dilution of species R1 can be parametrised, respectively, as:

% Vd/BLHEIGHT : R1 = ;
% DILUTE : R1 = ;

where Vd is the deposition velocity in cm s-1 , BLHEIGHT is the boundary layer height in cm and DILUTE is the dilution rate in s-1. BLHEIGHT and DILUTE are environment variables (Sect. 2.3), and therefore can be set to a value chosen by the user or constrained to prescribed values."

AtChem does not explicitly include entrainment of stratospheric air into the troposphere, although this process can be easily parametrised in the same way as the emissions can be parametrised. Note that the description of non-chemical processes outlined above is rather simplistic, although it is good enough for several applications. Implementation of more sophisticated approaches to model non-chemical processes is left to the users, according to their specific needs.

The choice of examples shown to demonstrate the AtChem system are at odds with the description of why it was developed. For example line 345 "AtChem2 was developed specifically for the long and complex simulations need for field studies". AtChem2 is then demonstrated using only 2 days from a ~40 day TEXAQs campaign. Were these two days chosen because they provided the best model-observation comparison? Why not show the whole TEXAQs time series, which would show that AtChem has been rigorously tested?

We chose to show only 3 days of the TexAQS model simulation mainly for reasons of clarity and simplicity. The entire model run, and associated analysis, is shown in Sommariva et al., (2011) and, although it was performed with an older version of AtChem, the agreement between the two versions is good (see line 359 and the modified Figure 7). We don't think that showing the whole time series would add much to this paper, which is a technical manuscript within the "Model description papers" remit of GMD.

It is true that the example in Section 3.2 is meant to show how AtChem2 can be used to carry out long simulations. However, please keep in mind that a fully constrained model, such as the one used for the TexAQS campaign, runs almost in real time. Depending on the computer system, a 3 days simulation takes between 2 and 4 days to complete. This is much longer than the chamber simulation shown in Section 3.1, which was completed in a couple of hours. Therefore we feel that the example shown in Section 3.2 provides an adequate demonstration of the enhanced capabilities of AtChem2.

**Editorial comments**

Title. I think GMD prefers a version number to be assigned to the model being described.

The version number has been added to the title (see also reply to the Editor's comments).

line 174 represents the start of a new paragraph but mentions 'this' modeling technique, which is not defined, or must relate to the paragraph above. I assume the latter, in which case it isn't a new paragraph.

The line break has been removed.

Line 204. 'study' not studies

Corrected.

Line 268. "since the model...' This and the following sentence are both very long, and could be broken up.

The paragraph has been rewritten as:

"The model configuration and constraints are read by the executable at runtime: there is no need to compile the model more than once, unless the chemical mechanism or parts of the source code are modified by the user (Fig. 5). This approach makes it quick and easy to set up batch model runs. With AtChem-online batch model runs are not possible because compilation is automatically performed on the web server when the model run is started: the chemical mechanism, the configuration files and the model constraints have to be uploaded via the Web Interface before every run and the model has to be recompiled every time it is executed, regardless of the changes that the user has made."

Line 350. Why is the latest version of the MCM not being used here?

The reason is that we wanted to be consistent with the model results published in Sommariva et al. (2011). See also the reply to a similar comment by referee #2.

Line 371. The reader needs to know what compounds were being constrained here – for example I'm assuming NO was constrained because of the statement about NO being the main destruction term. Earlier in this section reference is given to the model set-up in Sommariva et al (Sommariva et al., 2011), but as the results here rely upon the constraints they should be stated.

The sentence at line 353 has been rewritten as:

"The model constraints – CO, $CH_4$ , $H_2$ , NO, $NO_2$ , $O_3$ , $SO_2$ , $H_2O$, 65 VOC, j(O1D), j(NO2), j(NO3), aerosol surface area, temperature, pressure, latitude and longitude – and configuration were the same as in the model described by Sommariva et al. (2011)."

Figure 2. the plots are too small to be seen properly.

The size of the figure has been increased.

Figure 6, top right panel. A legend is missing for the three colors.

The legend has been corrected.

---

## Author Comment (AC3) · 19 Nov 2019

**1 General comments**

The paper describes a new open-source box model designed primarily for runs using the Master Chemical Mechanism (MCM), although other mechanisms can also be modelled provided they are entered in the correct format. The model comes in two forms, one (the original) available online via a web-interface, and a downloadable one (AtChem2) suitable for more advanced studies including batch runs that can be executed on the user's machine. The box model is presented as being free and easy to use, and having functionality that makes it particularly suitable for comparison against observational data, due to the way it handles constrained variables with different time frequencies. It is applied in two case studies to chamber and field data respectively to illustrate that it can handle these scenarios and allow us to reach some conclusions about which parts of the mechanism need to be reviewed. I believe this paper will be suitable for publication subject to revision.

We thank the referee for their comments and suggestions. Please find below our replies and the related modifications to the manuscript. The line numbers refer to the version of the manuscript published on GMDD.

**2 Specific comments**

The text in the paper is generally well-written and easy to follow. The figures are less clear, however. In particular, Figure 2 is too small to make much sense of even when zoomed in on (due to the scale); the outlier marker is larger than some of the box-and-whiskers. Figure 3 is missing some arrowheads, and the ones that are there are too small to be clearly seen. In Figure 6, the ozone and nitrogen oxides lines/markers are not labelled as to which is which; I would also recommend using different marker shapes when presenting multiple datasets in one graph in addition to using different colours.

The sizes of Figure 2 and 6 have been increased.
The arrows in Figure 3 have been fixed.
Markers have been added to Figure 6 and the legend has been corrected.

I think the paper could highlight in detail what makes AtChem stand out over specific other box models. The two forms (online/offline) appear to have different unique benefits, and perhaps need to be discussed separately. It appears that the online model can at present only be run with login details (is it only for use by the EUROCHAMP community?), which might restrict takeup (especially for teaching purposes), but I hope this is something the authors are planning to address. I also note that the online form does not seem to readily support the newest MCM version; perhaps this can be updated?

The referee is correct that the two versions of AtChem presented in this paper have different objectives, as well as different benefits and limitations (please note that the login details are available to anybody upon request, as indicated on the AtChem-online website, at https://atchem.leeds.ac.uk/webapp/). The history and the purpose of the two versions of AtChem are explained in the Introduction and in Section 2.1; we have also added more information about other open-source models in the Introduction (see the reply to a comment by referee #1).

Both versions of AtChem can be used with any version of the MCM, or with any general set of chemical reactions, as long as they are in the prescribed format. In fact, the examples in Section 3 were run with two different versions of the MCM (see below).

Given the commonality of the codebase we don't think it makes sense to divide the paper into two parts, each discussing a version of AtChem, as this would result in much repetition. We feel that the features and the characteristics that distinguish the two versions of the model are clearly identified as such in the manuscript.

Parts of the paper are using different versions of the MCM and perhaps also different versions of AtChem(2). In particular, on page 12 the section on lines 358-360 describes that the previously published results were off by a small amount due to a bug in a previous version of AtChem; however, at no point does the paper make reference to which version(s) of AtChem(2) were used in the runs. I would suggest carrying out all the model runs with the latest version of AtChem2 and annotating them as such. If the latest MCM version was not used for modelling the field study data for consistency with the previous paper analysing this data, I would comment on this, but also perhaps rerun the simulation with the newest MCM version for comparison.

All the model results discussed in Section 3 have been run with the same version of AtChem (version 1). The sentence on page 12, lines 358-360 refers to the model results published in Sommariva et al. (2011), not to those presented in this manuscript. The following changes have been made to the manuscript to clarify these points:

Lines 71-72: changed to "This paper presents version 1 of AtChem, and is divided into two parts"

Line 78: added "(version 1.5, rev. 146)"

Line 92: added " Version 1.0 of AtChem2 (doi:10.5281/zenodo.3404021) is presented here, and has been used for the model simulations shown in Sect. 3."

Line 358: changed to "The results obtained with version 1 of AtChem2 and with the beta version of AtChem used by Sommariva et al. (2011a) differ by ~3%"

It is true that different versions of the MCM are used in the paper. Specifically version 3.3.1 was used in Section 3.1 (chamber study example) and version 3.1 was used in Section 3.2 (field campaign example); the reason for this is that we wanted the model used in this paper to be directly comparable with the model used by Sommariva et al. (2011), in which the full model/measurement time series is presented.

For the objectives of this paper, the comparison between different versions of AtChem is more relevant than the comparison between different versions of the MCM, which is extensively covered in the updates to the original MCM protocol (e.g., see Saunders et al., 2003; Jenkin et al., 2015). If the same version of the MCM is used with the same configuration/constraints, then any difference between the two models can only be due to changes in the AtChem code (which was the case, see lines 358-360). To make this point clearer, Figure 7 was modified by adding the model results from Sommariva et al. (2011) and the text was modified as follows:

Lines 350-351: "The chemical mechanism used here was extracted from the MCM v3.1 (as in Sommariva et al. (2011a)): it included the inorganic chemistry scheme, the oxidation mechanism of 65 VOC, the dimethyl sulfide (DMS) oxidation mechanism from Sommariva et al. (2009), plus dry deposition terms and heterogeneous reactions for the appropriate gas-phase species."

Lines 354-356: moved to line 349.

**3 Technical corrections**

p7, line 204: change "studies" to "study"

Corrected.

p7, line 214: change "results" to "result" and correct spelling of "stiffness"

Corrected.

p8, line 224: Eq. 3 appears to use two different forms of the multiplication sign

Corrected.

p12, line 351: IUPAC spelling is "sulfide"

Corrected.

p22, label on Figure 4: the local time is GMT–7 in the winter, but GMT–6 in the summer due to daylight saving time

The plot is showing the correct times in GMT. The caption of Figure 4 has been corrected as suggested.

p27, in Table 1: "SO2" should have 2 as a subscript.

Corrected.

There is inconsistent use of American and British spellings (s/z) in the manuscript, and though it is a very minor issue, this can perhaps be standardised. I also feel that a few more hyphens would aid easy parsing of the text in places (e.g. p7, line 205 as well as Figure 2 label: "9-days"; p9, line 250 as well as p8, line 226: "two-stream").

The spelling has been corrected to British English. The hyphens have been added, as suggested.